# Off the shelf: Investigating transfer of learning using commercially available virtual reality equipment

Logan T. Markwell [1,2] *, Koleton Cochran[3], Jared M. Porter[1]

1 Department of Kinesiology, Recreation and Sport Studies, The University of Tennessee, Knoxville, Tennessee, United States of America, 2 Department of Physical Education and Health in Biała Podlaska, Józef Piłsudski University of Physical Education in Warsaw, Biała Podlaska, Poland, 3 Vetern Affairs Physical Therapy Clinic, Iowa City, Iowa, United States of America

* logan.markwell@awf.edu.pl, loganmarkwell95@gmail.com

**Data Availability Statement:** Data are contained within the manuscript and/or supporting information files.

**Funding:** The author(s) received no specific funding for this work.

## Abstract

The evolution of virtual reality (VR) has created the opportunity for a relatively low-cost and accessible method to practice motor skills. Previous studies have demonstrated how motor skill practice in non-immersive virtual environments transfers to physical environments. Though minimal research has investigated motor learning and transfer within immersive VR, multiple experiments provide empirical evidence of positive transfer effects. Most studies have used software engines and modified hardware to enhance the similarities between virtual and physical environments. However, many learners and practitioners currently use commercially available VR intending to enhance real-world performance, though there is very little evidence to support the notion of positive transfer for these systems. Therefore, this experiment aimed to investigate how motor skill practice using a commercially available VR system improved real-world performance and how that compared to physical practice. Participants (n = 68) were randomly selected into one of two groups: virtual reality (VR) practice (n = 33) or real-world (RW) practice (n = 35). The experiment occurred over two consecutive days, with a pretest, posttest, and practice phase occurring each day. The pre- and post-testing phases were identical for both groups and consisted of putting a golf ball 10 times on a carpeted surface towards the center of a target. The practice phases consisted of 60 total putts per day in the respective environment (VR or RW). Participants continuously alternated golf putting holes from three different distances until they accrued 60 total putts. Participants in the RW group performed golf putts to three targets. Participants in the VR group also performed golf putts on three different miniature golf putting holes, using the commercially available Oculus Rift and the Cloudlands VR Minigolf game. The VR putting targets were designed to replicate the putting holes in the physical environment. Separate 2 (group) x 4 (test) repeated measures ANOVAs were used to assess accuracy and club head kinematics. The results revealed a significant main effect for test, but not for group. Post hoc analyses revealed that both groups significantly improved their putting accuracy and club head kinematics at similar rates. The results from this study indicate that the transfer of learning that occurred as a result of practicing in a commercially available VR environment was similarly effective when compared to RW practice.

**Competing interests:** The authors have declared that no competing interests exist.

## Introduction

Virtual reality (VR) has recently gained popularity as a method for motor skill development [1]. Consider a surgeon who needs to practice a specific suture, a baseball player who needs to practice hitting a knuckleball, or a pilot who needs access to a helicopter and finances to cover the cost of flying the aircraft. The evolution of immersive VR (e.g., VR systems in which the user wears a head mount display and is fully encapsulated within the virtual environment) has created the potential for a relatively low-cost method of practice that is commercially available. Furthermore, one of the largest benefits of VR is the ability to instantaneously adapt to the challenge level based on the individual's skill level, maximizing learning potential [2]. Scientific investigations have found various benefits for using VR as a form of training [2–6]. However, research remains scarce examining how skills learned in an immersive virtual environment transfer out of the virtual space and into a physical, real-world (RW) environment. The results have been mixed among the studies investigating the effect of transfer of learning from immersive VR to a RW environment [e.g., 7–10]. For example, multiple experiments provided evidence that practicing a motor skill in immersive VR can increase performance within a RW environment [8 experiment 2,9,10]. However, other experiments have not supported this conclusion [7,8 experiment 1]. The studies above have used a variety of methods. As a result, the precise factors that contribute to transfer of learning from VR to the real world need to be clarified through additional research.

Interestingly, most studies investigating this topic have used VR technology that utilizes customized software or software engines designed to alter the virtual environment (e.g., Unity; Unreal). For example, the Unity Experiment Framework (UXF) developed by Brookes et al. [11] has been used during human behavior research using VR [see 8 for an example]. UXF allows the researcher to modify independent variables and create highly controlled testing conditions within the virtual environment that are nearly identical to the RW environment, increasing the physical and psychological similarities between the two realities (i.e., VR & RW). In addition to customized software, the hardware is frequently modified to further enhance the physical similarities between the VR and real-world tasks. For example, Harris et al. [8] used a physical golf club and attached sensors to create a VR club. Similarly, Oagaz et al. [10] used a VR table tennis racket with a similar size, weight, and shape to a physical racket. These software customizations and hardware modifications have provided obvious methodological benefits for testing real-world performance improvements through the use of VR. However, in a practical setting, it is unlikely that all practitioners (e.g., physical therapist, coach, flight instructor) or learners (e.g., patients, athletes, students) will possess the needed skills to program a software engine (e.g., UXF) that can modify the virtual environment to simulate the RW environment with high fidelity. Likewise, not all users will have modified hardware (e.g., controller attachments) for VR training. Instead, some will ultimately use "off-the-shelf" commercially available applications and hand-held controllers. The lack of physical and psychological similarities when using non-customized commercially available VR hardware and software might alter the extent to which transfer of learning occurs from a virtual to a real-world setting.

Several explanations have been proposed to explain transfer of learning. For example, the identical elements theory [12], later evolved into Singley and Andersons's [13] identical production model, and the transfer-appropriate processing theory [14] are traditional motor learning theories that have dominated the literature. These theories propose that to achieve positive transfer of learning, similarities must exist between the practice and transfer conditions [e.g., practice specificity; 15]. However, the identical elements theory posits that positive transfer is due to the similarities between the movement characteristics (e.g., the swing of a

golf club) and/or the environmental context in which the skill is performed [12]. On the other hand, the transfer-appropriate processing theory suggests that transfer occurs due to cognitive processing similarities [14]. The cognitive process similarities can be considered the type or amount of information the individual must process within the practice environment (e.g., intrinsic & extrinsic feedback). Evidence from testing the two explanations suggests both have merit [13, 14]. Thus, it can be expected that the degree to which positive transfer of learning occurs is related to the shared similarities of the skill characteristics, environmental context, and cognitive processes. Such explanations also align with the practice specificity literature [15]. Specifically, research testing the predictions of the practice specificity hypothesis suggests that if sources of information available during the learning phase of a skill are removed, performance is likely to deteriorate [15, 16]. Given that many software and hardware companies are marketing VR systems as a method to enhance real-world performance, understanding whether these devices can be taken off the shelf and used without modifications and what VR factors may or may not contribute to a positive transfer of learning is imperative to investigate from both a practical and theoretical perspective.

To our knowledge, only two studies have examined the transfer of learning effects from a virtual environment to a RW environment using non-customized, commercially available immersive VR hardware and software [7, 9]. Michalski et al. [9] investigated how table tennis practice in immersive VR compared to a no-practice control group. Participants performed a total of three hours and 30 minutes of table tennis using a readily available application. Their analysis showed that the VR group significantly outperformed the control group, suggesting that immersive VR was beneficial for improving performance compared to no practice at all. More recently, Drew et al. [7] compared performance and kinematic differences between dart-throwing practice in immersive VR and the real world. Both groups completed blocks of 10 dart throws until they accrued a total of 100 throws. The VR group used a commercially available application and a hand-held controller during the practice session. Drew et al. [7] demonstrated that dart-throwing accuracy significantly decreased following practice in VR. In contrast, accuracy increased following RW practice, as evidenced by a posttest in the RW environment immediately following practice. Additionally, the results also showed that there were kinematic differences between the groups during practice. However, no kinematic differences existed during the posttest, suggesting that both practice groups used similar movements to accomplish the task during the posttest. Therefore, though both practice groups led to similar movement characteristics during the posttest, RW dart throwing not only outperformed VR dart throwing during the real-world posttest, but contrary to Michalski et al. [9], practice in VR led to worse real-world performance compared to the pretest. In other words, the findings reported by Drew et al. [7] demonstrated that practicing a motor skill in VR hindered performance. This decrease in performance likely occurred due to the lack of similarities between conditions. This study [7] and others [1] highlight the necessity of understanding how VR transfers to real-world performance.

The limited evidence examining the effects of RW performance improvements using commercially available VR is mixed, and it is unclear whether VR practice is as effective as RW practice. Specifically, to our knowledge, Drew et al. [7] is the only study that has compared VR practice, without software or hardware customizations, to RW practice of the same task. Therefore, the purpose of the present study was to examine the transfer of learning effects using commercially available VR software and hardware during practice. Using a golf putting task, we compared real-world accuracy and club swing kinematics after VR or RW practice. Though a VR hand-held controller was used, we predicted that both forms of practice would elicit accuracy improvements due to the similarities between the movement and environment characteristics and the cognitive processes involved. This prediction was based on our

understanding of the transfer of learning effects [13–15] and previous studies that provided initial evidence of positive transfer [8 experiment 2,9,10]. Additionally, we hypothesized that analyses of the kinematics during the posttests would reveal similarities between groups, consistent with previous research [7].

## Method

### Participants

Participants (n = 68) were recruited from undergraduate kinesiology classes to participate in this study. The participants were informed that they would practice a golf putting task but were naïve to the purpose of the study. All participants read and signed an informed consent prior to participation, and all forms and methods were approved by Southern Illinois University's Institutional Review Board.

### Apparatus and task

The data collected for this experiment took place in a climate-controlled research laboratory. A golf putting task was used for both groups (VR practice; RW practice). A golf putting task was used because it is a commonly used task to investigate motor learning, and therefore the number of trials performed in this study was expected to result in motor learning [8, 17, 18]. Additionally, given that the purpose of the study was to assess a commercially available application, golf putting was a task that could be performed using a publicly available VR application. Lastly, a golf putting task was used as it provides logistical benefits when recreating a laboratory-based RW condition. The RW practice consisted of putting golf balls toward targets at three different distances (.91 m, 1.37 m, 1.83 m) on a carpeted surface inside a climate-controlled research laboratory. Participants used a standard length (90 cm) golf putter for all pre- and post-testing trials to putt a regular-sized (diameter 4.27 cm) golf ball towards a target. The target was a series of concentric circles. The center circle had a diameter of 10.8 cm, and each concentric ring had a diameter that increased by 10.8 cm, respectively. The concentric ring in which the ball came to rest determined the score for each putt. The center circle resulted in a score of zero, a score of one was recorded if the ball came to rest in the next circle, and so on for each respective ring out to a 15th circle. A score of 16 was recorded if the ball came to rest outside of the last ring. If the ball came to rest on a line of any ring, the participant received a score for the innermost ring.

The Oculus Rift VR headset and the Cloudlands VR Minigolf application were used to create three virtual golf putting holes designed to replicate the putting holes in the RW environment by shape and length. Participants used a virtual golf putter and ball to putt into a virtual hole while wearing the Oculus Rift headset and holding one Oculus controller in their dominant hand.

A digital camera, with a capture rate of 60 hz, was placed perpendicular to the participant at a distance of two meters. The digital camera recorded the action of the golf putt. Dartfish software was used to measure the distance of the backswing and follow-through displacement. The time-course measure in the Dartfish software was used to determine the amount of time it took participants to complete each putt. Golf club head velocity was calculated by determining how quickly the club head moved through the range of motion from the maximum backswing position to ball contact for each putt.

### Procedure

Participants were randomly assigned into one of two groups: VR practice (n = 33) or RW practice (n = 35). After participants signed the consent form, the researcher provided instructions

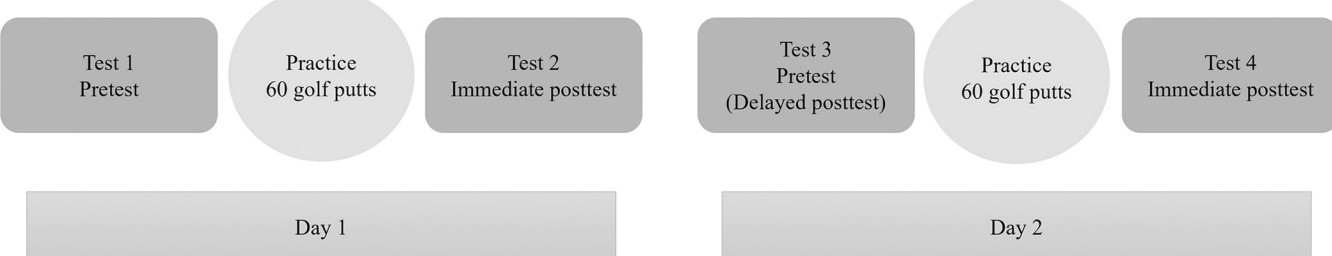

**Fig 1. Schematic representation of the experimental procedures used during days one and two of the experiment.**

followed by a demonstration of the golf putting task. The participant was instructed to hit the ball onto the center of the target or as close to the center of the target as possible.

The experiment took place over two consecutive days, with a pretest, practice, and posttest phases occurring on both days (see Fig 1). The pre- and post-test phases were identical for both groups. During the pre- and post-testing phases, participants putted a golf ball 10 times on the carpeted surface toward the center of the target from a distance of 1.83 m. The practice phases consisted of 20 golf putts from three different distances, totaling 60 putts within the respective environment (i.e., RW or VR). During the practice phases, participants continuously alternated golf putting holes (i.e., a,b,c, a,b,c . . .) from all three distances (i.e., .91 m, 1.37 m, 1.83 m) until they had accrued 60 total putts for each day, regardless of the group. A significant amount of motor learning research has provided evidence that variable serial practice (i.e., a,b,c . . .) results in larger motor learning improvements compared to constant practice (i.e., a,a,a . . .) [e.g., 19, 20]. Thus, a serial practice design was used to enhance the motor learning effects from both practice groups. Participants returned 24 hours later (i.e., day 2) to complete a pretest (i.e., delayed posttest), which also served as a delayed retention test for the RW group or a transfer test for the VR group. Following the day 2 pretest, a practice phase of 60 total putts and a posttest were performed. Day one and day two were identical in structure. At the end of the second day, participants accrued 120 total putts across both practice phases.

## Statistical analysis

There were four separate 2 (group) x 4 (test) repeated measures analysis of variance (ANOVAs) used to assess accuracy, backswing displacement, follow-through displacement, and club velocity differences between groups and across tests. The homogeneity of variance was examined using Levene's test. Sphericity was examined using Mauchly's test of sphericity. When significant, degrees of freedom were corrected using Greenhouse-Geisser estimates.

## Results

### Performance outcome variable

**Accuracy.** A 2 (group) x 4 (test) repeated measures analysis of variance (ANOVA) was used to determine accuracy differences between groups and tests. Mauchly's test indicated that the assumption of sphericity was violated ($\chi^2(5) = 19.227$, $p = .002$). Therefore, degrees of freedom were corrected using Greenhouse-Geisser estimates of sphericity ($\varepsilon = .828$). The analysis revealed a significant main effect for test $F(2.485, 156.574) = 5.693$, $p = .002$, $\eta_p^2 = .083$. The interaction between the test and group was not significant, $p = .298$. Furthermore, the test of between-subject effects revealed a non-significant effect, $p = .660$. In light of the significant main effect for test, pairwise comparisons were made to determine differences across tests (see

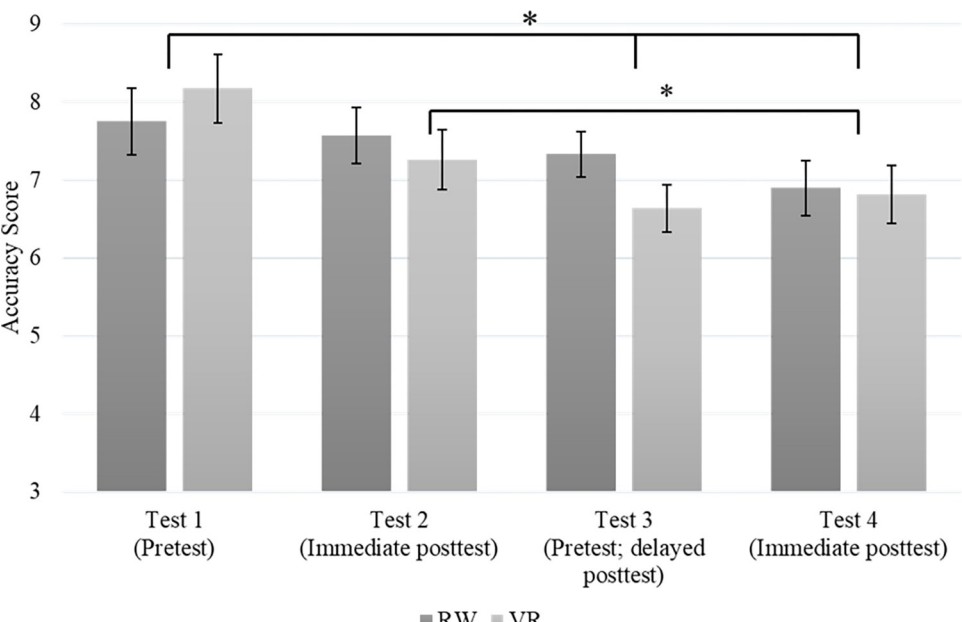

**Fig 2. Accuracy score differences between groups and across tests.** * p = < 0.05.

Fig 2). The analysis revealed that test three ($M$ = 6.986, $SD$ = 1.698) was significantly lower compared to test one ($M$ = 7.964, $SD$ = 2.436), $p$ = .006. The analysis also revealed that test four ($M$ = 6.857, $SD$ = 2.050) was significantly lower compared to test one, $p$ = .002. Additionally, the analysis revealed that test four ($M$ = 6.857, $SD$ = 2.050) was significantly lower compared to test two ($M$ = 7.418, $SD$ = 2.108), $p$ = .029.

### Performance production (Kinematic) variables

**Club backswing displacement.** A 2 (group) x 4 (test) repeated measures analysis of variance (ANOVA) was used to determine kinematic differences during club backswing between groups and tests. Mauchly's test indicated that the assumption of sphericity was violated ($\chi^2(5)$ = 34.318, $p < .001$), therefore degrees of freedom were corrected using Greenhouse-Geisser estimates of sphericity ($\varepsilon$ = .719). No significant differences were found between groups, $p$ = .737, or within-subjects across tests, $p$ = .086.

**Club follow-through displacement.** A 2 (group) x 4 (test) repeated measures analysis of variance (ANOVA) was used to determine kinematic differences during club follow-through between groups and tests. Mauchly's test indicated that the assumption of sphericity was violated ($\chi^2(5)$ = 13.525, $p < .019$), therefore degrees of freedom were corrected using Greenhouse-Geisser estimates of sphericity ($\varepsilon$ = .862). The analysis revealed a significant main effect for test $F(2.587, 162.959)$ = 4.842, $p$ = .005, $\eta_p^2$ = .071. The interaction between the test and group was not significant, $p$ = .475. Additionally, the test of between-subject effects revealed a non-significant effect, $p$ = .107. Given the significant main effect for test, pairwise comparisons were made to determine differences across tests (see Fig 3). The analysis revealed that the displacement during test two ($M$ = .196, $SD$ = .071) was significantly larger compared to test one ($M$ = .176, $SD$ = .063), $p < .001$. The analysis also revealed that the displacement during test four ($M$ = .191, $SD$ = .067) was significantly larger than test one ($M$ = .176, $SD$ = .063), $p$ = .006. Furthermore, comparisons showed that the displacement during test three ($M$ = .181, $SD$ = .065) was significantly lower compared to test two ($M$ = .196, $SD$ = .071), $p$ = .005.

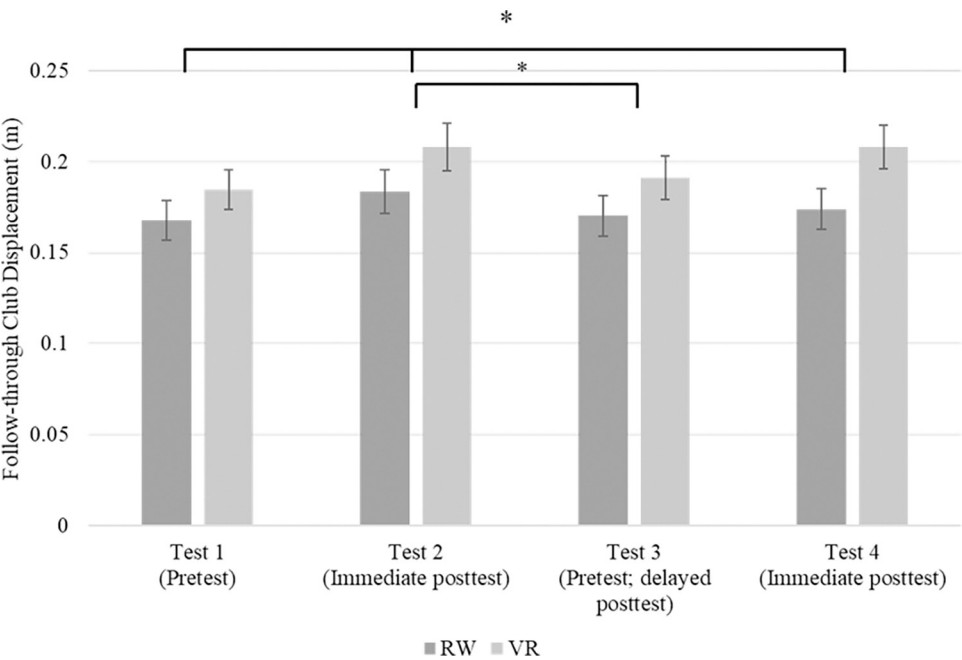

**Fig 3. Follow-through displacement differences across tests.** * p = < 0.05.

**Club velocity.** A 2 (condition) x 4 (test) repeated measures analysis of variance (ANOVA) was used to determine club velocity differences between groups and tests. The analysis revealed a significant main effect for test $F(3, 183) = 2.682$, $p = .048$, $\eta_p^2 = .042$. No significant interaction between test and group was found, $p = .735$. Additionally, the test of between-subject effects revealed a non-significant effect, $p = .051$. Considering the significant main effect, pairwise comparisons for test were made to determine differences across tests (see Fig 4). The

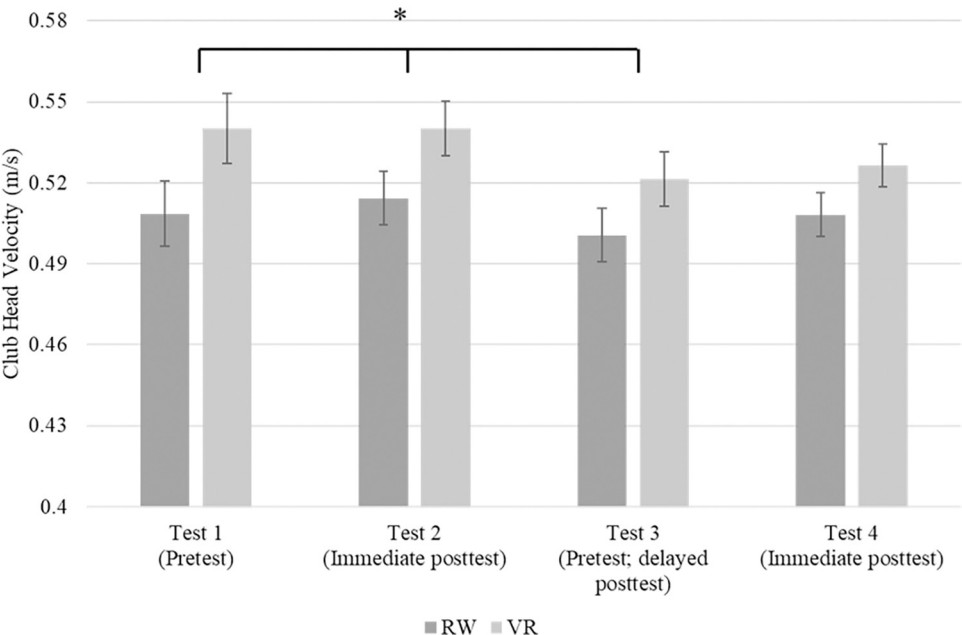

**Fig 4. Club head velocity differences across tests.** * p = < 0.05.

comparisons revealed club velocity during test three (*M* = .511, *SD* = .055) was significantly lower compared to test one (*M* = .524, *SD* = .071), *p* = .048 and test two (*M* = .527, *SD* = .057), *p* = .012.

## Discussion

The purpose of this study was to investigate the real-world golf putting accuracy and kinematic differences between RW and VR practice using a commercially available device and application. Based on previous research demonstrating positive transfer of learning following VR practice [9] and established theoretical frameworks [13–15], we predicted that both groups would facilitate motor learning, but no statistical differences would be observed between groups. Based on Drew et al.'s [7] findings, we also hypothesized that there would not be group differences across the club swing kinematic measures. As predicted, the analyses of the kinematic data revealed that the practice of putting a golf ball in VR or the RW resulted in similar changes in RW golf swing kinematics. Also, in line with our predictions, both groups significantly improved golf putting accuracy at similar rates, and no differences were observed between the groups. In other words, swinging a virtual club to hit a virtual ball towards a virtual target resulted in accuracy and club swing technique improvements similar to swinging a real golf club to hit a real golf ball towards a real target. These results indicate that practicing a motor skill using commercially available VR technology can lead to similar RW performance improvements and biomechanical similarities when compared to RW practice.

The observations in the present experiment support the predictions made by the identical element's theory [12]. Specifically, this theory predicts that similarities between movement and context characteristics will result in positive transfer of learning. The data in this study showed that the VR group shared similar movement characteristics with the RW group, resulting in similar motor learning effects. Thus, the congruent findings between the performance production (i.e., kinematics) and performance outcome (i.e., accuracy) measurements support predictions made by the identical element's theory [12]. Furthermore, the present findings also support the transfer-appropriate processing theory [14]. This theory purports that the more cognitive processing similarities (e.g., decision-making, problem-solving, application of rules, amount of interference, type, and amount of feedback) shared between two learning environments, the greater the amount of positive transfer of learning should occur [14]. Since the same task, with the same rules, and with a similar amount of variability, was performed in both practice environments, it is not surprising to see that the present study's results support the predictions made by the transfer-appropriate processing theory [14]. Thus, the current experiment supports both theories [12, 14].

The present study adds to the small body of research examining motor skill transfer within immersive VR. More so, this is only one of a few studies investigating the efficacy of using commercially available VR systems to improve RW motor learning. Michalski et al. [9] were one of the first to establish that commercially available VR training is beneficial compared to no training at all. Here, and in line with Michalski et al. [9], the present study demonstrates that practicing with commercially available VR systems can improve performance and appear to be similarly effective compared to RW practice. Furthermore, these results also align with previous studies that have investigated the transfer of learning for both immersive [8 experiment 2,9,10] and non-immersive VR technology [2, 21–23].

It is important to note that our findings are inconsistent with a recent study conducted by Drew et al. [7], in which they found that dart-throwing practice using an HTC Vive, hand-held controllers, and a commercially available application led to a decrease in real-world dart throwing performance. This negative transfer of learning likely occurred due to differences

between the task performed in VR compared to the RW. Specifically, the virtual dart board height was scaled to the participant's height in the virtual environment. In contrast, the dart board height in the RW environment was standard at a fixed height across all participants regardless of the person's height. These task and environment differences reduced the similarities between conditions and likely led to the observed decrease in performance [12, 13, 16]. In the present study, we scaled the virtual environment so the golf putting holes and putting distances were similar between the VR and RW conditions. Such similarities likely contributed to our observed positive transfer of learning.

Another primary difference between the present study and Drew et al. [7] was the methods used in the investigations. Specifically, the present study investigated performance differences from pre- to posttests *immediately* following the acquisition phase. Additionally, the present experiment examined the performance differences during a *delayed* posttest, with a minimum of 24 hours between the end of the acquisition phase on day one and the delayed posttest, which occurred on day two. In contrast, Drew et al. [7] only examined performance differences *immediately* following the acquisition phase. Interestingly, during the present study, no performance improvements were observed between the pretest and the immediate posttest on the first day of testing. Instead, performance improvements from the pretest were only observed on the delayed posttest and the second day's immediate posttest. The performance improvements observed on the second day, but not the first, are likely due to memory consolidation [24]. Such findings highlight the importance of following well-established recommendations [19, 20] for providing a minimum of one day between the end of the acquisition phase and the delayed posttest (i.e., retention or transfer) when investigating the motor learning effects of practice, including acquisition in VR environments.

In the present experiment, practice in both groups led to similar biomechanical measurements. Specifically, there were no club head displacement or velocity group differences in the posttest following VR or RW practice. While very few studies have examined performance production measures, such as biomechanical kinematics [1], a positive transfer within biomechanical measurements has been observed in both non-immersive [21] and immersive VR [7]. The present results revealed that both groups similarly increased club head follow-through displacement and decreased club head velocity due to practice. A longer follow-through has been shown to be a characteristic of skilled putters [25, 26]. Therefore, the increased follow-through displacement observed in our study could have been due to increased skill level following the practice session. Regarding the observed decrease in velocity, the speed-accuracy tradeoff (i.e., Fitts' law) suggests that individuals tend to exchange speed to maintain or increase levels of accuracy [27]. Thus, it is unsurprising to have observed a decrease in club velocity as individuals increased golf putting accuracy. Other golf putting research has indicated similar findings that a lower club head velocity is optimal for higher levels of accuracy [28, 29].

To our knowledge, this is one of the first studies to show that practicing a motor skill using commercially available VR leads to similar outcomes and mechanics compared to physical practice. While these findings provide theoretical and practical value, this study is not without limitations. First, the empirical evidence testing transfer of learning using immersive VR technology is still relatively scarce. While this study provides evidence that commercially available VR can effectively lead to performance improvements in a real-world setting, much is still to be discovered regarding how these transfer of learning results generalize to other hardware and software configurations. Furthermore, future research should continue to investigate how the practice of other types of motor skills transfer to the RW environment in comparison to RW practice. Additionally, studies should investigate the extent to which the level of fidelity between virtual and physical realities can differ while still achieving a positive transfer of

learning effect. Another limitation is the proximity to statistical significance (p = 0.051) observed in the present study for the club velocity between-group analysis. Club velocity may have reached a statistical difference between groups in a larger sample size. Importantly, however, the mean differences between the groups decreased as the amount of practice increased. Extending the number of practice trials and days to investigate how extended amounts of practice influence performance production and outcome measures would provide valuable insight. Lastly, although not common in motor learning research, future investigations should evaluate learning effects following VR practice over greater periods of time (e.g., one week, one month) following the cessation of acquisition to examine the permanency of observed improvements.

Despite these limitations, the results of the present study suggest that motor skill practice using a commercially available VR application and hardware can effectively elicit a positive transfer of learning to a real-world environment. More specifically, using a readily available golf putting application and hand-held controllers produced biomechanical and accuracy improvements that did not statistically differ compared to golf putting in the real world. These findings provide insights for practitioners in sport, military, clinical, and industry settings. Specifically, the results from the present study suggest that commercially available VR can offer an effective alternative or supplement to traditional skill acquisition methods. This has been of specific interest to clinical [e.g., 30–32] and industry [e.g., 33] domains for rehabilitation and training purposes, respectively. However, as with any developing technology, understanding the generalizability of the motor learning effects from VR is important. Before using VR as a form of practice or training, validating that the technology does indeed elicit a positive transfer of learning is crucial for the given context.

## Supporting information

**S1 Data.**
(XLSX)

## Author Contributions

**Conceptualization:** Jared M. Porter.

**Data curation:** Logan T. Markwell, Koleton Cochran.

**Formal analysis:** Logan T. Markwell.

**Supervision:** Jared M. Porter.

**Writing – original draft:** Logan T. Markwell.

**Writing – review & editing:** Logan T. Markwell, Jared M. Porter.

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
