## [Decision Letter · Decision Letter 0]

7 Jun 2023

PONE-D-22-34509Off the shelf: Investigating transfer of learning using commercially available virtual reality equipmentPLOS ONE

Dear Dr. Markwell,

Thank you for submitting your manuscript to PLOS ONE. After careful consideration, we feel that it has merit but does not fully meet PLOS ONE’s publication criteria as it currently stands. Therefore, we invite you to submit a revised version of the manuscript that addresses the points raised during the review process.

We look forward to receiving your revised manuscript.

Kind regards,

Nick Fogt

Academic Editor

PLOS ONE

Journal Requirements:

2. Please provide additional details regarding ethical approval in the body of your manuscript. In the Methods section, please ensure that you have specified the name of the IRB/ethics committee that approved your study.

3. Please ensure that you include a title page within your main document. You should list all authors and all affiliations as per our author instructions and clearly indicate the corresponding author.

Additional Editor Comments:

This paper addresses an important question regarding virtual reality training and motor transfer. Of particular concern here is Reviewer #1s and Reviewer #2s comments regarding the statistical analyses and the appropriateness of the study design in terms of directly addressing the question of transfer the authors sought to answer. These comments from both reviewers should be carefully addressed. The authors should also comment on the advantages of Reviewer #1s proposal of analyzing changes (perhaps on an individual basis) within the training environment (eg. virtual training with outcomes in virtual reality) in addition to the transfer to real-world conditions. In other words, is there something to be learned by examining the changes that occur within each environment? Of perhaps most concern is Reviewer #1s comment on the time course of the study. Can the authors provide some reasoning or evidence as to whether such a short period of training and testing is adequate to support the authors' conclusions?

Lastly, the introduction should be edited for content and length as suggested by Reviewer #1, and the discussion should center around how the results relate to theories (eg. identical elements theory) discussed in the introduction.

Reviewers' comments:

Reviewer's Responses to Questions

**Comments to the Author**

1. Is the manuscript technically sound, and do the data support the conclusions?

Reviewer #1: No

Reviewer #2: Yes

2. Has the statistical analysis been performed appropriately and rigorously? 

Reviewer #1: No

Reviewer #2: Yes

3. Have the authors made all data underlying the findings in their manuscript fully available?

Reviewer #1: Yes

Reviewer #2: Yes

4. Is the manuscript presented in an intelligible fashion and written in standard English?

Reviewer #1: No

Reviewer #2: Yes

5. Review Comments to the Author

Reviewer #1: This is an interesting article regarding motor learning and transfer, however, there are some concerns related to the organization of the manuscript, methods and statistical analisys. Which made me assume, that this kind of work need to be better strutured.

1. Extremely extensive introduction, with moments that should be placed in the discussion sector and not introduction.

I believe that better organization of this part is necessary, for a more fluid and better organized line of reasoning on the subject in question. Due to the amount of information, it becomes tiring, extensive and the central objective of the work is confusing.

2. Statistical analysis, poorly organized and poorly described.

How was the homogeneity test performed?

How was the data distribution verified so that the ANOVA could be performed?

What were the kinetic variables evaluated?

The information is scattered throughout the text, in the section where only the results should be written.

3. The article presents problems of statistical analysis and methods that are concerning. If transfer was what was being analyzed, it was not necessary to perform a repeated measures analysis considering interaction factors between and within groups. Only a pre and post test analysis comparing the practice groups was enough.

And actually, what is being compared are the differences between practice groups and not the transfer of learning itself.

Also, it takes more practice time and more retention analysis time to verify that the individual has really learned. Remembering that learning translates into continuous changes in behavior over time, with at least 24 hours of a good night's sleep so that there is retention and, after this the transfer proposal.

Information about the study participants is not clearly found, which further mades more bias for the data.

A more correct pilot study, to later carry out a transfer study, would be with two practice groups: virtual reality and real world, and the practice observations of each group, with the learning curves of each individual.

The virtual reality group practices and is analyzed through virtual reality.

The real world group practices and is analyzed by the real world.

The learning curves are verified and then comparisons between groups from the learning results, taking into account the practice.

And later on, a proposal for learning transfer.

It is not even known correctly whether this practice of golf in virtual reality is good enough to be compared with the real world.

4. Which is the importance of this practice? Why was golf chosen? This is not clear.

Reviewer #2: In this article the authors sought to investigate whether training a mechanical task (i.e., swinging a golf club for putting) in virtual reality (VR) transfers to the real world (RW) and how this virtual training compares to real-world training. The authors designed and empirical study in which the experiment took place over two days. On the first day, participants were tasked with completing a pre-test in which they putted a golf ball 10 times in the real world, followed by a 60 practice putts in either a VR or RW training session, followed by a post-test task of 10 additional putts in the real world. This procedure was then repeated on the second day. Data was recorded for accurary of the putts, as well as kinematic measurse of club backswing, follow-through and velocity. The authors anticipated that both groups would improve in putting performance, with no significant difference between the VR and RW training groups. Further, they anticipated that there would be no significant difference between the two groups in terms of kinematic performances. The data appear to generally support these hypotheses.

Overall, this paper adds to the limitted existing literature that examines how commercially available VR programs can be used for training mechanical tasks. These findings are important as they provide additional evidence that VR training may be comparable to real-world training. Given the wide range of diciplines that are attempting to use VR for training purposes, this research is both very relevant and impactful. Recommendation is to accept, provided authors address some minor issues.

Introduction

• While the previous work of Michalski et al. and Drew et al. were included, it appears that the respresentation of Drew et al.’s findings are slightly misleading. In line 122-123 it is stated that Drew et al.’s findings did not demonstrate kinematic differences during post test; while correct, it should be noted that kinematic differences were observed during training, and that participants appeared to adjust their methods during the post-test task.

Method

• It is unclear why three distances were used for training (ln 180-181)? Did participants do the same number of putts at each distance, or did they choose where to putt from? Did the experimenters control for the potential of a recency effect of practicing at the 1.83 distance when the pre-test and post-test were just done at 1.83?

• The procedure of how kinematic data was measured is missing from the method section and should be provided for replicability purposes.

Figures and Tables

• The figure caption for Figure 1 suggests it is for general procedure but includes phrase "Delayed Transfer" under Test 3 which only relates to the RW condition and not the VR condition - either the figure or the caption should be adjusted to more accurately present what is being shown.

Results, Discussion & Conclusions:

• Pairwise comparisons were included in the results, but an explanation was not posited for the differences observed. Would be beneficial for these differences to be addressed (i.e. For accuracy scores, why both test 3 and 4 for significantly lower than test 1 but test 2 was not; would it both be expected that the immediate post-test would also demonstrate improvement of scores after the training session? Do the authors have a possible explanation for this pattern of results?)

• Without the specifics of how kinematic data was measured, it is difficult to interpret the results provided.

• In line 239 the authors provide a non-significant p-value of p = 0.051 and use this value as support for no between-group differences in club velocity. However, given the very close proximity to significance, and the lack of a power analysis provided, this value should be addressed in the discussion as what could be a possible group difference in larger samples and the implications.

• It would help to have the reported implications addressed in terms of the theoretical framework discussed in the introduction of the manuscript. Specifically, how do the results align (or not align) with the identical elements theory, identical production model and/or transfer appropriate processing theory? Do these theories offer potential explanation for why the observed results occurred?

• Limitations to the study design should be included in the discussion.

6. PLOS authors have the option to publish the peer review history of their article (what does this mean?). If published, this will include your full peer review and any attached files.

Reviewer #1: No

Reviewer #2: No

---

## [Author Response · Author response to Decision Letter 0]

3 Aug 2023

Each reviewer comment has been addressed and included within our "Response to Reviewers" file.

---

## [Decision Letter · Decision Letter 1]

25 Aug 2023

PONE-D-22-34509R1Off the shelf: Investigating transfer of learning using commercially available virtual reality equipmentPLOS ONE

Dear Dr. Markwell,

Thank you for submitting your manuscript to PLOS ONE. After careful consideration, we feel that it has merit but does not fully meet PLOS ONE’s publication criteria as it currently stands. Therefore, we invite you to submit a revised version of the manuscript that addresses the points raised during the review process.

We look forward to receiving your revised manuscript.

Kind regards,

Nick Fogt

Academic Editor

PLOS ONE

Journal Requirements:

Additional Editor Comments:

Both reviewers agreed that the author has done well responding to their comments.

Please briefly address the remaining few comments from reviewer #1. Specifically, address the need for longer-term follow-up in motor learning studies, and the applicability/practicality (or lack thereof) currently of using virtual reality training clinically.

Reviewers' comments:

Reviewer's Responses to Questions

**Comments to the Author**

1. If the authors have adequately addressed your comments raised in a previous round of review and you feel that this manuscript is now acceptable for publication, you may indicate that here to bypass the “Comments to the Author” section, enter your conflict of interest statement in the “Confidential to Editor” section, and submit your "Accept" recommendation.

Reviewer #1: All comments have been addressed

Reviewer #2: (No Response)

2. Is the manuscript technically sound, and do the data support the conclusions?

Reviewer #1: Yes

Reviewer #2: Yes

3. Has the statistical analysis been performed appropriately and rigorously? 

Reviewer #1: Yes

Reviewer #2: Yes

4. Have the authors made all data underlying the findings in their manuscript fully available?

Reviewer #1: Yes

Reviewer #2: (No Response)

5. Is the manuscript presented in an intelligible fashion and written in standard English?

Reviewer #1: Yes

Reviewer #2: Yes

6. Review Comments to the Author

Reviewer #1: The authors provided viable and better-founded responses to all queries. In addition, they proposed a reorganization of writing and demonstrating the data with greater clarity.

For complete acceptance, I suggest that the authors place a study limitation related to the non-longitudinal follow-up (greater than one month follow-up) of the observed performance gains. Also that, new studies need to do a longer follow-up to verify greater gains in motor learning.

I also suggest placing a small paragraph in the discussion, related to the clinical applicability of training in virtual reality, taking into account why this training will be efficient.

After thet inclusions on discussion section, I think the article can be accept.

Reviewer #2: (No Response)

7. PLOS authors have the option to publish the peer review history of their article (what does this mean?). If published, this will include your full peer review and any attached files.

Reviewer #1: No

Reviewer #2: No

---

## [Author Response · Author response to Decision Letter 1]

19 Sep 2023

The responses to reviewers have been included as an attached file. Each comment has been addressed.

---

## [Editor Report · Decision Letter 2]

21 Sep 2023

Off the shelf: Investigating transfer of learning using commercially available virtual reality equipment

PONE-D-22-34509R2

Dear Dr. Markwell,

We’re pleased to inform you that your manuscript has been judged scientifically suitable for publication and will be formally accepted for publication once it meets all outstanding technical requirements.

Kind regards,

Nick Fogt

Academic Editor

PLOS ONE

Additional Editor Comments (optional):

Thank you for addressing all of the editorial and reviewer comments.
---

## [Editor Report · Acceptance letter]

25 Sep 2023

PONE-D-22-34509R2 

Off the shelf: Investigating transfer of learning using commercially available virtual reality equipment 

Dear Dr. Markwell:

I'm pleased to inform you that your manuscript has been deemed suitable for publication in PLOS ONE. Congratulations! Your manuscript is now with our production department. 

Kind regards, 

on behalf of

Dr. Nick Fogt 

Academic Editor

PLOS ONE